# Screening and Identification of Anti-Inflammatory Compounds from Erdong Gao via Multiple-Target-Cell Extraction Coupled with HPLC-Q-TOF-MS/MS and Their Structure–Activity Relationship

**DOI:** 10.3390/molecules28010295

**Published:** 2022-12-30

**Authors:** Mengyu Li, Hui Luo, Zhen Huang, Jin Qi, Boyang Yu

**Affiliations:** Jiangsu Key Laboratory of TCM Evaluation and Translational Research, School of Traditional Chinese Pharmacy, China Pharmaceutical University, Nanjing 211198, China

**Keywords:** Erdong Gao, anti-inflammatory, steroidal saponins, structure activity relationship, cell extraction

## Abstract

Erdong Gao (EDG), consisting equally of roots of Asparagi Radix and Ophiopogonis Radix, is a well-known traditional Chinese formulation that has been used to treat cough and throat pain for centuries. However, the bioactive components in EDG remain to be elucidated. In this study, a rapid and effective method involving live cell bio-specific extraction and HPLC-Q-TOF-MS/MS was established to rapidly screen and identify the anti-inflammatory compounds of an EDG extract. One hundred and twenty-four components were identified in EDG extract using HPLC-Q-TOF-MS/MS analysis. After co-incubation with 16HBE, HPAEpiCs and HUVECs, which have been validated as the key target cells for pulmonary diseases, sixteen components were demonstrated to exhibit an affinity for binding to them. Furthermore, fifteen components were subsequently verified to exert anti-inflammatory effects on lipopolysaccharide (LPS)-induced 16HBE, HPAEpiCs and HUVECs via inhibiting the release of TNF-α and IL-6, indicating that nine steroidal saponins may possess potential for the treatment of lung-related diseases. Taken together, our study provides evidence that live cell biospecific extraction combined with the HPLC-Q-TOF-MS/MS technique was an efficient method for rapid screening potential bioactive components in traditional Chinese medicines and the structure activity relationship of steroidal saponins in EDG was summarized for the first time.

## 1. Introduction

Erdong Gao (EDG), which is composed of the roots of *Asparagus cochinchinensis* (Lour.) Merr. and *Ophiopogon japonicus* (Linn. f.) Ker-Gawl. (1:1, *w*/*w*), was recorded in *She Sheng Mi Pou* written by Hong Ji of the Ming Dynasty [1]. This formulation nourishes the yin and moistens the lung, and has been used to prevent and treat pulmonary diseases; such as fever and cough in clinics for hundreds of years. Previous studies have shown that steroidal saponins and homoisoflavones are the major components of this formulation, including ophiopogonin D, ophiopogonin D′, dioscin, methylprotodioscin, officinalisnin-II, aspacochinosides N, aspacochinosides O, aspacochinosides P, ophiopogonanone D and Methylophiopogonone B, etc. [2,3,4,5,6,7,8,9,10,11], which displays significant anti-inflammatory, antibacterial, antioxidant and immunomodulatory activities [12,13,14,15,16,17,18]. EDG modulates inflammatory responses and has been employed to prevent and treat pulmonary diseases; such as acute lung injury (ALI) by reducing pulmonary edema, and the secretion of TNF-*α* and IL-6, and increasing the expression of aquaporin-1 (AQP-1) and aquaporin-5 (AQP-5) proteins in the lung tissue [19]; it also markedly reduces the expression of TNF-*α*, interleukin-10 (IL-10), and transforming growth factor-β1 (TGF-β1) and promotes the apoptosis of gastric cancer cells to delay the occurrence and development of lung tumors in rats [20]. EDG containing serum played an anti-cancer role by inhibiting the proliferation, migration and invasion in A549 cells [21]. In brief, EDG is mainly used to treat pulmonary diseases, which effects on these diseases are directly related to anti-inflammatory activity. However, the anti-inflammatory activity components of EDG are unclear and need to be further investigated.

Pulmonary diseases are often associated with infiltration of inflammatory cells and elevated responses to inflammatory proteins such as cytokines, resulting in damage of internal organs and acute respiratory stress syndrome [22]. Lipopolysaccharide (LPS), a main component of the outer membranes of Gram-negative bacteria, has been identified as a key risk factor for pulmonary diseases, such as acute lung injury (ALI), acute respiratory distress syndrome (ARDS) and chronic obstructive pulmonary disease (COPD) [23,24]. Researchers found that when LPS was injected into the body, the levels of pro-inflammatory cytokines increased [25]. Inflammation is the first host immune response to develop to protect the body from injury or infection, and mediated by pro-inflammatory cytokines including TNF-α, IL-6, IL-1 and IL-8 [26,27]. In addition, the airway epithelium is the most important tissues in the pathogenesis of respiratory diseases and, by expressing different inflammatory genes, plays an important role in disease aggravations [28], and the alveolar epithelium and vascular endothelium are important components of the lung barrier and blood-air barrier, prompting rapid production of proinflammatory cytokines [29]. Therefore, airway epithelium cells, alveolar epithelium cells and vascular endothelium cells have been widely used in research on pulmonary diseases and therapeutic drugs or measures as target cells [30], which play a crucial role during inflammation by regulating immune responses.

Traditional Chinese medicines (TCMs) are characterized by multiple chemical components, multiple targets and multiple effects. Compound isolation, identification/structure elucidation, and resupply are necessary to exploit the potential of individual bioactive compounds for drug development [31]. However, the traditional screening methods based on extensive isolation and subsequent pharmacological evaluation of the isolates are not only time-consuming and laborious, but also prone to missing minor but vital bioactive molecules [32,33,34]. Recently, living cell extraction coupled with analytical techniques has been successfully developed to screen in the complex system of TCMs for bioactive components [35,36,37,38,39], which is based on the hypothesis that interaction with receptors or channels on the cell membrane or transportation into the inner cell is required for a compound to elicit a biological effect [40,41]. Additionally, this method has several advantages; it can be used to extract active components from complex samples with high specificity, does not require complicated procedures for preparing bio-chromatographic columns, and allows identification of the extracted compounds by HPLC-MS/MS. Although this method is rapid and convenient, it needs to be tested and verified for its ability to screen bound components. Therefore, we first identified the components of EDG using HPLC-Q-TOF-MS/MS. Secondly, multiple-target-cell extraction (16HBE, HPAEpiCs and HUVECs) coupled with HPLC-Q-TOF-MS/MS was conducted to screen and determine the potential bioactive components in EDG. Moreover, the anti-inflammatory activities of fifteen components extracted in this manner were verified by LPS-induced injury in target cells. Eventually, nine steroidal saponins were proved to be the main anti-inflammatory active components of EDG, and their structure–activity relationship was summarized. 

## 2. Results 

### 2.1. The Identification of Components in EDG via HPLC-Q-TOF-MS/MS Analysis

Steroidal saponins and homoisoflavonoids were reported as the major ingredients in EDG. In order to facilitate the detection of steroidal saponins, especially furostanol saponins, HPLC-ELSD was used to separate the chromatographic peaks in EDG. In addition, considering the maximum absorption wavelength of homoisoflavonoids, 290 nm was selected for the experiment [42]. Detection of chromatographic peaks was followed by HPLC-Q-TOF MS/MS analysis to identify components. The major components in EDG were identified or tentatively characterized via comparisons of the retention behaviors and mass fragment ion information with those in the previous references and those of available reference compounds [43,44,45,46,47,48,49]. Finally, one hundred and twenty-four compounds, including ninety-seven steroidal saponins and twenty-two homoisoflavonoids were identified, which as shown in Figure 1 and Appendix A.

### 2.2. Effects of EDG on LPS-Induced Injury in Target Cells

To investigate the effects of EDG on inflammatory cytokine secretion in vitro, the levels of TNF-*α* and IL-6 in LPS-stimulated 16HBE, HPAEpiCs and HUVECs were measured using ELISA kits. Target cells were incubated with EDG at concentrations of 5–50 μg/mL for 12 h, and cell viabilities were measured (Figure 2a–c). As shown in Figure 2a,c, when the extract was used at a concentration of 50 μg/mL, cytotoxicity was observed in 16HBE and HUVECs; Figure 2b shows that at an extract concentration ranging from 30 to 50 μg/mL, cytotoxicity was still seen in HPAEpiCs. Therefore, a concentration range of 5–20 μg/mL was selected for the final dosage. As shown in Figure 2d–f, LPS treatment markedly increased TNF-α levels compared with the control treatment (*p* < 0.001), and treatment with EDG significantly inhibited the production of TNF-*α* as compared with the LPS group (*p* < 0.05 or *p* < 0.001). Similarly, a significant decrease in the levels of IL-6 was observed in three target cells (*p* < 0.05 or *p* < 0.001); the results are shown in Figure 2d–f. The results of three target cells showed that EDG had a marked anti-inflammatory capability in a dose-dependency. Thus, it was of our interest to identify the components underlying the observed anti-inflammatory activity of this EDG extract.

### 2.3. Screening of Potential Anti-Inflammatory Components in EDG Extracts

According to the established in vitro model, target cells were divided into three groups: the LPS+EDG group, the LPS group and the control group. After pretreatment, the cells were washed six times to remove non-selectively bound ingredients and collected. The eluate was collected for HPLC analysis, and for comparison, PBS was also washed six times. Then, all eluates were discarded except for the last one, which, together with LPS+EDG, LPS and control samples, was collected for HPLC-MS analysis. The chromatograms are shown in Figure 3, via extracting the total ion chromatogram and conducting an intercomparison, sixteen compounds were identified in 16HBE cells, seventeen compounds were identified in HPAEpiCs, and twenty compounds were identified in HUVECs (Table 1). Finally, fifteen components were found to bind to all three cell types and their chemical structures are shown in Figure 4.

The steroidal saponins in EDG that bound to target cells were mainly spirostane-type disaccharides and trisaccharides. The aglycones included sarsasapogenin, ruscogenin, diosgenin, ophiopogon, pennogenin and pragerigenin A. The sugar included glucose, galactose, rhamnose, xylose, etc. In addition, homoisoflavones also bound to target cells. After analysis, we found that the polarity of the compounds and the position and quantity of the substituent groups have a greater impact on its combination with the target cells. Therefore, we speculated that the fifteen components that bound to the three target cells were the potential anti-inflammatory compounds in EDG.

### 2.4. Effects of the Potential Active EDG Components on LPS-Induced Injury in Target Cells 

Despite live cell extraction with high probability and high specificity, which could be used to extract active components from complex samples, the bound components still needed to be verified. Thus, the effects of ED1–15 were evaluated in LPS-induced injury in 16HBE, HPAEpiCs and HUVECs; 16HBE, HPAEpiCs and HUVECs were incubated with ED1–15 at concentrations of 0.01–10 μM for 12 h and cell viability was measured (Appendix A). ED1–ED7 at 10 μM showed significant cytotoxicity. Thus, the concentrations of 0.01, 0.1 and 1 μM were selected as the final dosages.

As shown in Figure 5 and Figure 6, 16HBE, LPS treatment significantly increased the levels of TNF-*α* and IL-6 compared with those in the control group (*p* < 0.001), and treatment with ED6 and ED7 (0.01 μM) significantly reduced the production of TNF-*α* in 16HBE cells as compared with the LPS group (*p* < 0.05). Moreover, a significant decrease in the levels of IL-6 was observed upon treatment with ED1 and ED7 (0.01 μM) in this group as compared with the LPS group (*p* < 0.05). The levels of TNF-*α* and IL-6 were found to be remarkably descended upon treatment with ED1, ED6, ED7, ED8, and ED9 (0.1 μM) in this group as compared with the LPS group (*p* < 0.05 or *p* < 0.01). Interestingly, treatment with ED1-10 (1 μM) resulted in a significant decline in the secretion of TNF-*α* and IL-6. However, after treatment with ED11-ED15 (0.01–1 μM), the levels of TNF-*α* and IL-6 typically showed a decreasing trend.

Similarly, as demonstrated in Figure 5 and Figure 6, a significant increase in the levels of TNF-*α* and IL-6 was observed upon LPS-induced injury in HPAEpiCs as compared with the control group (*p* < 0.001), and after treatment with ED1–15 (0.01 μM), the levels of TNF-*α* and IL-6 generally showed a decreasing trend (*p* > 0.05). The levels of TNF-*α* and IL-6 were found to decline significantly (*p* < 0.05 or *p* < 0.01) upon treatment with ED1, ED2, ED6, ED7 and ED10 (0.1 μM). Interestingly, treatment with ED1-10 (1 μM) resulted in a significant decline in the secretion of TNF-*α* and IL-6, except for ED5. Nevertheless, the levels of TNF-α and IL-6 commonly showed a decreasing trend after administration of ED11-15 (0.01–1 μM); this result is consistent with 16HBE cells. 

Likewise, as shown in Figure 5 and Figure 6, LPS treatment significantly increased TNF-α and IL-6 levels compared with control group (*p* < 0.001) in HUVECs, and a generally decreasing trend of TNF-*α* and IL-6was observed after treatment with ED1–15 (0.01 μM). The levels of TNF-*α* and IL-6 were found to decline significantly (*p* < 0.05 or *p* < 0.01) upon treatment with ED1, ED2, ED3, ED6, ED7, ED8, ED9 and ED10 (0.1 μM) in comparison to the LPS group, and treatment with ED1-10 (1 μM) resulted in the significant reversal of TNF-*α* and IL-6 levels toward a normal level (*p* < 0.01). However, the levels of TNF-α and IL-6 generally showed a decreasing trend after administration of ED11-15 (0.01–1 μM); this result is consistent with 16HBE and HPAEpiCs.

In conclusion, the results for the release of TNF-α and IL-6 in the three target cells showed that the activities of steroidal saponins were better than those of homoisoflavones, and the activities of spirostanol saponins were better than those of furostanol saponins. By analyzing the chemical structure of spirostanol saponins, it was found that spirostanol saponins with ruscogenin and sarsasapogenin as the mother nucleus and carbonyl substitution at position C-12 had stronger anti-inflammatory activity than other types, which indicates that the type of saponin and the type and position of substituents in steroidal saponins are important factors affecting their activities. This result is consistent with the reported results [48,49]. Thus, steroidal saponins, especially spirostanol saponins, might be potential anti-inflammatory components in EDG. In addition, the anti-inflammatory activity of steroidal saponins was closely related to the type of sapogenin, the position of substituent and sugar moiety substitution. 

## 3. Discussion

HPLC-Q-TOF-MS/MS technology was employed to identify the components of EDG. one hundred and twenty-four compounds, including ninety-seven saponins and twenty-two homoisoflavonoids were identified. The homoisoflavonoids and saponins which the aglycone contain sarsasapogenin, ruscogenin, ophiopogon, pennogenin and pragerigenin A are come from *O. japonicus*. Additionally, most of the saponins with diosgenin originate from *A. cochinchinensis*. In addition, both positive and negative modes yielded abundant information for structural identification, and the negative ion mode was chosen for further analysis because most of the compounds in EDG contain functional groups such as hydroxyl, glycosyl, etc., and most of components can be detected in the negative ion mode. Due to the factors such as the ionization of compounds, the type of eluting solvent and the adsorption of chromatographic column, there are still some components in EDG that cannot be completely identified, and further research is needed.

Furthermore, multiple-target-cell extraction combined with the HPLC-Q-TOF-MS/MS technique was employed for the screening and identification of potential anti-inflammatory components in EDG, showing that fifteen components bind to the three target cells. For steroidal saponins that uncombine to target cells, are usually furostanol saponins or acetyl or thiol groups form the sugar chain contains, and for homoisoflavones that uncombine to target cells, no substitution was observed at the C-8 position or methoxy substitution at the C-6 position. The results illustrate that the size of compounds as well as the position and type of substituents are important factors affect affinity for combination with target cells. In this study, multiple-target-cell extraction combined with the HPLC-Q-TOF-MS/MS technique was used but some potentially bioactive compounds might not be detected by this method. In addition, with other washing steps or conditions, some compounds might be washed away, or other compounds might have been bound to the cells, and some bioactive compounds might be missed during analysis.

Finally, the activities of fifteen potential anti-inflammatory components in LPS-induced injury in target cells were evaluated. The results are shown in Figure 5, Figure 6 and Figure 7, (1) Regarding the types of components, the anti-inflammatory activity of steroidal saponins was superior to that of homoisoflavones; (2) for different aglycones, spirostanol saponins with ruscogenin and sarsasapogenin as the parent nucleus and with a carbonyl substitution at position C-12 had better anti-inflammatory activity; (3) regarding the substitution position of the sugar chain, anti-inflammatory activity increased when there was a substitution at C-1 position, but its cytotoxicity also increased; (4) For spirostanol saponins that bound with target cells, the anti-inflammatory activities of disaccharides and trisaccharides at the C-3 position were similar; (5) For spirostanol saponins demonstrating desaturation between the C-5 and C-6 positions, the anti-inflammatory activity did not change significantly; (6) For spirostanol saponins with hydroxyl substitution at the C-13 and C-14 positions, the anti-inflammatory activity was similar [50,51]; (7) For steroidal saponins bound with target cells, the anti-inflammatory activity of spirosterol saponins was superior to that of furostanol saponins. The above results suggest that steroidal saponins exhibited stronger anti-inflammatory activity than homoisoflavones and the activity of spirosterol saponins was superior to that of furostanol saponins, which may be related to the polarity and substituents of the compounds.

## 4. Materials and Methods

### 4.1. Chemicals and Reagents

Acetonitrile and methanol for HPLC were obtained from Tedia (Fairfield, OH, USA); absolute ethanol was obtained from Shanghai Titan Scientific Co., Ltd. (Shanghai, China); Dulbecco’s Modified Eagle’s Medium (DMEM), fetal bovine serum, trypsin, penicillin, phosphate-buffer saline (PBS) and streptomycin were all purchased from GIBCO (Invitrogen Corporation, Carlsbad, CA, USA); 3-[4,5-Dimethylthiazol-2-yl]-2,5-diphenyltetrazolium bromide (MTT) was purchased from AMRESCO (Cleveland, OH, USA); dimethyl sulfoxide (DMSO) was obtained from Damao Chemical Reagent Factory (Tianjin, China); and Dexamethasone (Dex) was purchased from Aladdin (Shanghai, China). Commercial kits used for determining tumor necrosis factor-α (TNF-*α*) and interleukin-6 (IL-6) were purchased from Yifeixue Biotechnology Co., Ltd. (Nanjing, China). (25*R*)-3*β*-[(*O*-*β*-D-glucopyranosyl (1→4)-*β*-D-galactopyranosyl) oxy]-5*β*-spirostan-12-one, (25*S*)-3*β*-[(*O*-*α*-L-rhamnopyranosyl (1→2)-*β*-D-galactopyranosyl)oxy]-5*β*-spirostan-12-one, 3-*O*-*α*-L-rhamnopyranosyl(1→2)-[*β*-D-glucopyranosyl(1→4)]-*β*-D-glucopyranosyl(25 *S*)-5*β*-spirostan-3*β*-ol, (25*S*)-5β-spirostan-3*β*-ol-3-*O-β*-D-glucopyranosyl(1→2)-*β*-D-glucopyranoside, (25*R*)-*26-O-β*-D-glucopyranosyl-5-en-3*β*,22*α*,26-triol-furost-3-*O*-*α*-L-rhamnopyranosyl-(1→4)-*O*-[*α*-L-rhamnopyranosyl(1→2)]-*O*-*β*-D-glucopyranosid, Ophiopogonin D, Ophiopogonin D’, Cixi-ophiopogon C, pennogenin-3*-O-α*-L-rhamnopyranosyl-(1→2)-*β*-D-xylopyranosyl-(1→4)-*β*-D-glucopyranoside, prazerige-nin A-3*-O-α*-L-rhamnopyranosyl-(1→2)-*β*-D-glucopyranoside, Methylophiopogona-none A, Methylophiopogonanone B, Methylophiopogonone A, Ophiopogonanone E, Methylophiopogonone B were isolated in our laboratory (purity ≥ 95%) and their structures were identified by NMR [52]. All the chemicals and reagents were of analytical grade. All solvent and samples were filtered through 0.22 μm membranes before use.

### 4.2. Preparation of EDG

The roots of *Asparagus cochinchinensis* (Lour.) Merr. were collected from Yulin in the Guangxi Zhuang Autonomous Region (Yulin, China); the roots of *Ophiopogon japonicus* (Linn. f.) Ker-Gawl. were collected from Mianyang in Sichuan Province (Sichuan, China) and were authenticated by Professor Jin-Qi form the China Pharmaceutical University. Asparagi Radix and Ophiopogonis Radix (1:1, *w*/*w*) were crushed into powder and sieved, which powder (100 g) was soaked in water (1 L) at 30 °C for 1h and refluxed for extraction three times at 100 °C, the first time for 3 h, and the second and third times for 2 h, respectively [53]. The filtrate was concentrated under reduced pressure and subjected to chromatographic separation on a D101 macroporous adsorption resin column, eluted with water and 90% ethanol. The 90% ethanol eluate was collected, and concentration determination and lyophilization were performed to prepare a freeze-dried powder.

### 4.3. Cell Culture and Cell Extraction 

The 16HBE, HPAEpiCs and HUVECs were obtained from Shanghai Institute of Cell Biology, Chinese Academy of Sciences (Shanghai, China). The cells were cultured in DMEM with 15% FBS medium and maintained in 5% CO_2_ at 37 °C. Target cells were cultured in 25 cm dishes, and cells of each line were divided into three groups: control, LPS, and LPS+EDG. EDG was added when the cell density reached 90%. The EDG solution was removed after incubation for 4 h, cells were washed with PBS 6 times, and the eluate was collected. Then, 80% ethanol was added into the dish, followed by sonication at 4 °C for 20 cycles (30 s × 280 W) and centrifugation at 4 °C, 1000 rpm for 5 min. A termovap sample concentrator was used to concentrate the supernatant, and the residue was dissolved in methanol (200 μL). The samples were then filtered, and HPLC-Q-TOF-MS/MS was performed for analysis. 

### 4.4. Chromatography and Mass Spectrometry Conditions

Chromatography analysis was performed on an Agilent Series 1260 liquid chromatograph (Agilent Technologies, Santa Clara, CA, USA), equipped with an Alltech ELSD 3300 detector (GRACE Alltech, Columbia, MD, USA). A Waters X-bridge C18 column (4.6 mm × 250 mm, I.D., 5 μm, Serial No. 01943817913855) was used.The mobile phase A was water containing 0.1% formic acid and phase B was acetonitrile containing 0.1% formic acid. The gradient elution conditions were as follows: 0–15% B at 0–15 min; 22% B at 15–30 min; 22–24% B at 30–35 min; 24% B at 35–45 min; 24–29% B at 45–50 min; 29% B at 50–60 min; 29–48% B at 60–90 min; 48–52% B at 90–100 min; 52–65% B at 100–110 min; 65–95% B at 110–120 min. The flow rate was 1 mL/min, and the column temperature was set at 30 °C. The injection volume was 10 μL. The signal was acquired at 290 nm.

HPLC-Q-TOF-MS/MS analysis was performed on an Agilent series 1260 liquid chromatograph (Agilent Technologies) coupled with an Agilent 6530 Q-TOF mass spectrometer (Agilent Technologies) equipped with an ESI source. The mass parameters were optimized as follows: drying gas temperature, 350 °C; drying gas (N_2_) flow rate, 10.0 L/min; fragmentor voltage, 120 V; nebulizer gas pressure, 45 psi; capillary voltage, 3500 V; Oct RFV, 750 V; MS/MS collision energy was set at 15–70 V. Mass spectra were recorded in the range of *m*/*z* 100–1500 [42,54]. Each sample was analyzed in both positive and negative modes to give abundant information for structural identification. All the acquired data and analysis were processed using Mass Hunter software 10.0 (Agilent Technologies).

### 4.5. Cell Viability Assay

The 16HBE, HPAEpiCs and HUVECs growing exponentially were trypsinized and then approximately 3000 cells/well were seeded into 96-well plates. After the experimental treatment, cells were incubated with MTT at a final concentration of 0.5 mg/mL for 3 h at 37 °C. Then, the medium was discarded and 150 mL of DMSO was added to dissolve the formazan crystals. The absorbance was read at 570 nm with a reference wavelength of 650 nm, and cell viability was expressed as the percentage of absorbance to control values.

### 4.6. Enzyme-Linked Immunosorbent Assays (ELISAs)

ELISAs were performed to test the levels of TNF-α and IL-6 in supernatants using respective ELISA Kits (YiFeiXue Bio-technology) according to the manufacturer’s instructions. The absorbance was then determined at 450 nm using a microplate reader (BioTek, Winooski, VT, USA).

### 4.7. Statistical Analysis

All the experimental data were processed using Graph Pad software (San Diego, CA, USA) and expressed as the mean ± SEM. Statistical analysis was carried out using Dunnett’s test for comparison when the data involved three or more groups. Probability values *p* < 0.05 were defined as statistically significant.

## 5. Conclusions

Erdong Gao (EDG) is a famous TCM formulation for the treatment of pulmonary diseases in clinicals. However, there have been few reports on the anti-inflammatory component screening and analysis of EDG and even fewer in the structure–activity relationship of saponins. Consequently, one hundred and twenty-four components were presumed and characterized in EDG. The fifteen compounds showed a binding affinity for target cells, and ten steroidal saponins had verified anti-inflammatory properties and a demonstrated ability to protect target cells against LPS-induced injury via inhibiting TNF-α and IL-6. Meanwhile, the structure–activity relationship of steroidal saponins that bound to target cells was summarized for the first time. The results provide essential data for the material basis of anti-inflammatory activity of EDG, and the methods are applicable to the study of other TCM prescriptions.

## Figures and Tables

**Figure 1 molecules-28-00295-f001:**
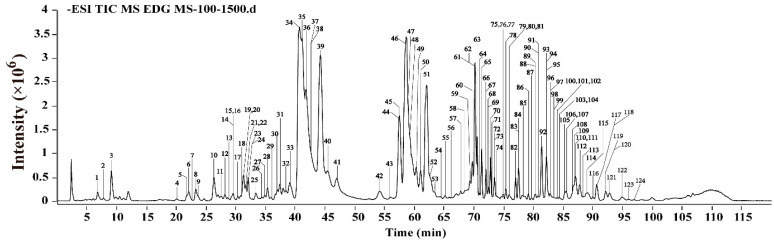
The total ion chromatograms (TICs) of EDG obtained with HPLC-Q-TOF/MS in negative ion mode.

**Figure 2 molecules-28-00295-f002:**
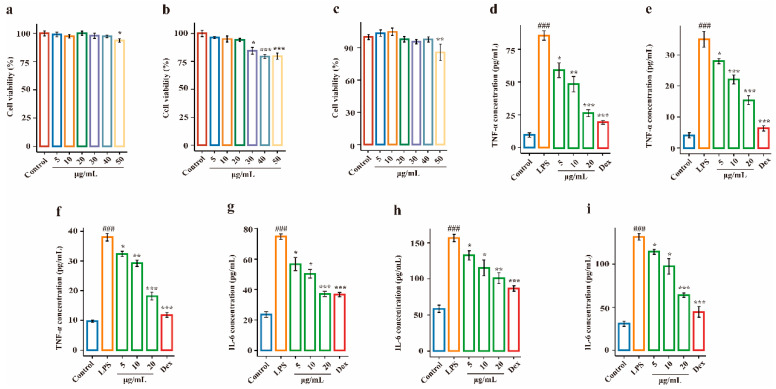
Effects of EDG on LPS-induced injury in target cells. (**a**–**c**) The cell viability of 16HBE, HPAEpiCs and HUVECs after treatment with different concentrations of EDG. (**d**–**f**) TNF-α production of 16HBE, HPAEpiCs and HUVECs, respectively. (**g**–**i**) IL-6 production of 16HBE, HPAEpiCs and HUVECs, respectively. The 16HBE, HPAEpiCs and HUVECs were pretreated with EDG at the concentration of 5–20 μg/mL for 12 h. Cell viability was measured with the MTT assay, and Dex was used as a positive control drug (5 μg/mL). TNF-*α* and IL-6 levels in the medium, as measured by ELISA. Results were obtained from three independent experiments and are presented as mean ± SEM. ^###^
*p* < 0.001 vs. Control; * *p* < 0.05, ** *p* < 0.01, *** *p* < 0.001 vs. LPS. LPS, Lipopolysaccharide; Dex, Dexamethasone.

**Figure 3 molecules-28-00295-f003:**
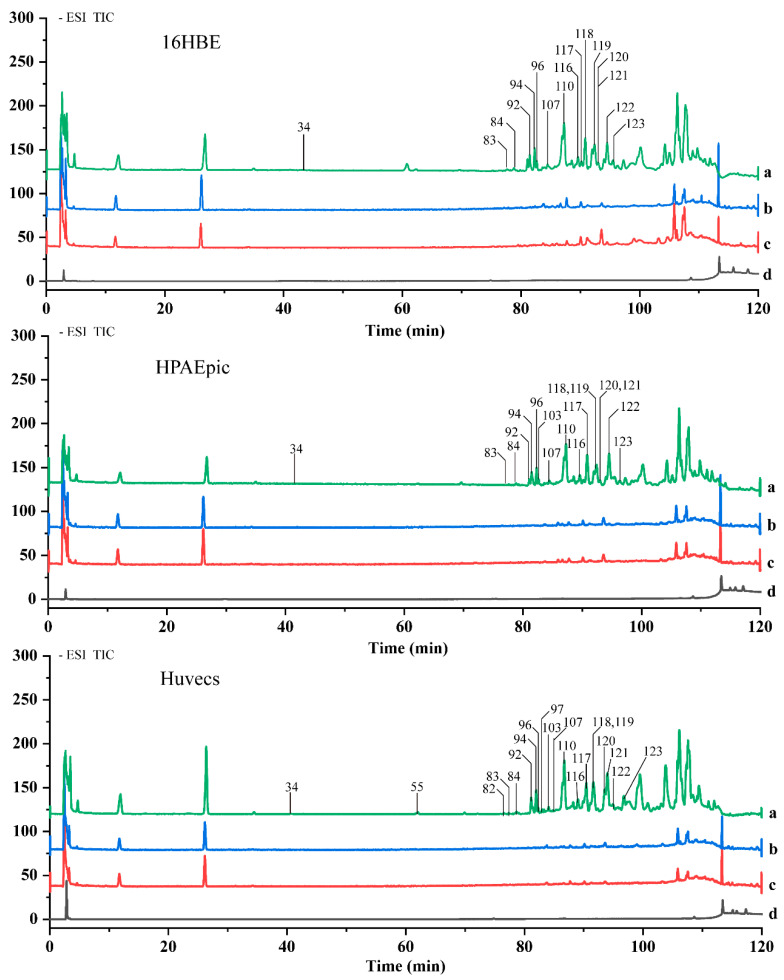
The total ion chromatograms (TICs) of EDG 16HBE, HPAEpiCs and HUVECs-binding molecules obtained with HPLC-Q-TOF/MS in negative ion mode. a: LPS+EDG group, target cells cultured with LPS (5 μg/mL) and EDG extraction (20 μg/mL); b: LPS group, target cells cultured with LPS (5 μg/mL); c: control group, target cells cultured with serum-free medium; d: final washing eluate group. P34: 26-*O*-*β*-D-glucopyranosyl-(25*R*)-5*β*-furostane-3*β*,22*α*,26-triol-3-*O*-*β*-D-xylopyranosy-l-(1→4)-*β*-D-glucopyranosyl(1→2)-*β*-D-glucopyranoside, P55: Ruscogenin 3-*O*-*α*-L-rhamnopy-ranosyl (1→2)-[*β*-D-glucopyranosyl(1→3)]-*O*-*β*-D-glucopyranoside, P82: Ophiopogenin-3-*O*-*α*-L-rham-npyranosyl-(1→2)-*β*-D-xylopyranosyl-(1→3)-*β*-D-glucopyranoside, P83: Cixi-ophiopog-on C, P84: (25*R*)-3*β*-[(*O*-*β*-D-glucopyranosyl-(1→4)-*β*-D-galactopyranosyl)oxy]-5*β*-spirostan-12-one, P92: Pennogenin 3-*O*-*α*-L-rhamnopyranosyl-(1→2)-*β*-D-xylopyranosyl-(1→4)-*β*-D-glucopy-ranoside, P94: Prazerigenin A 3-*O*-*α*-L-rhamnopyranosyl-(1→2)-*β*-D-glucopyranoside, P96: Ophiopogonanone E, P97: Prazerigenin A-3-*O*-*α*-L-rhamnopyranosyl-(1→3)-*β*-D-xylopyranosyl-(1→4)-*β*-D-glucopyranoside, P103: 5,7,2’-Trihydroxy-3’,4’-methylenedioxy-6,8-dimethyl homois-oflavone, P107: (25*S*)-3*β*-[(*O*-*α*-L-rhamnopyranosyl-(1→2)-*β*-D-galactopyranosyl)oxy]-5*β*-spiros-tan-12-one, P110: 3-*O*-*α*-L-rhamnopyranosyl-(1→2)-[*β*-D-glucopyranosyl(1→4)]-*β*-D-glucopyra-nosyl-(25*S*)-5*β*-spirostan-3*β*-ol, P116: (25*S*)-5*β*-spirostan-3*β*-ol-3-*O*-*β*-D-glucopyranosyl-(1→2)-*β*-D-glucopyranoside, P117: Ophiopogonin D, P118: Methylophiopogonone A, P119: Ophiopogonin D’, P120: Methylophiopogonone B, P121: Methylophiopogonanone A, P122: Methylophiopogonanone B, P123: Ophiopogonin B.

**Figure 4 molecules-28-00295-f004:**
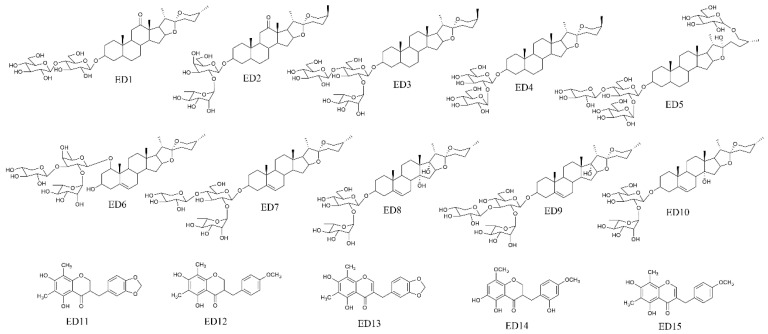
The chemical structures of the compounds that bound to 16HBE, HPAEpiCs and HUVECs. ED1: (25*R*)-3*β*-[(*O*-*β*-D-glucopyranosyl (1→4)-*β*-D-galactopyranosyl) oxy]-5*β*-spirostan-12-one, ED2: (25*S*)-3*β*-[(*O*-*α*-L-rhamnopyranosyl (1→2)-*β*-D-galactopyranosyl)oxy]-5β-spirostan-12-one, ED3: 3-*O*-*α*-L-rhamnopyranosyl(1→2)-[*β*-D-glucopyranosyl(1→4)]-*β*-D-glucopyranosyl(25*S*)-5*β*-spirostan-3*β*-ol, ED4: (25*S*)-5*β*-spirostan-3*β*-ol-3-*O-β*-D-glucopyranosyl(1→2)-*β*-D-glucopyranoside, ED-5: (25*R*)-*26-O-β*-D-glucopyranosyl-5-en-3*β*,22*α*,26-triol-furost-3-*O*-*α*-L-rhamnopyranosyl-(1→4)-*O*-[*α*-L-rhamnopyranosyl(1→2)]-*O*-*β*-D-glucopyranosid, ED6: Ophiopogonin D, ED7: Ophi-opogonin D’, ED8: Cixi-ophiopogon C, ED9: pennogenin-3*-O-α*-L-rhamnopyranosyl-(1→2)-*β*-D-xylopyranosyl-(1→4)-*β*-D-glucopyranoside, ED10: prazerigenin A-3*-O-α*-L-rhamnopyranosyl-(1→2)-*β*-D-glucopyranoside, ED11: Methylophiopogonanone A, ED12: Methylophiopogonanone B, ED13: Methylophiopogonone A, ED14: Ophiopogonanone E, ED15: Methylophiopogonone B.

**Figure 5 molecules-28-00295-f005:**
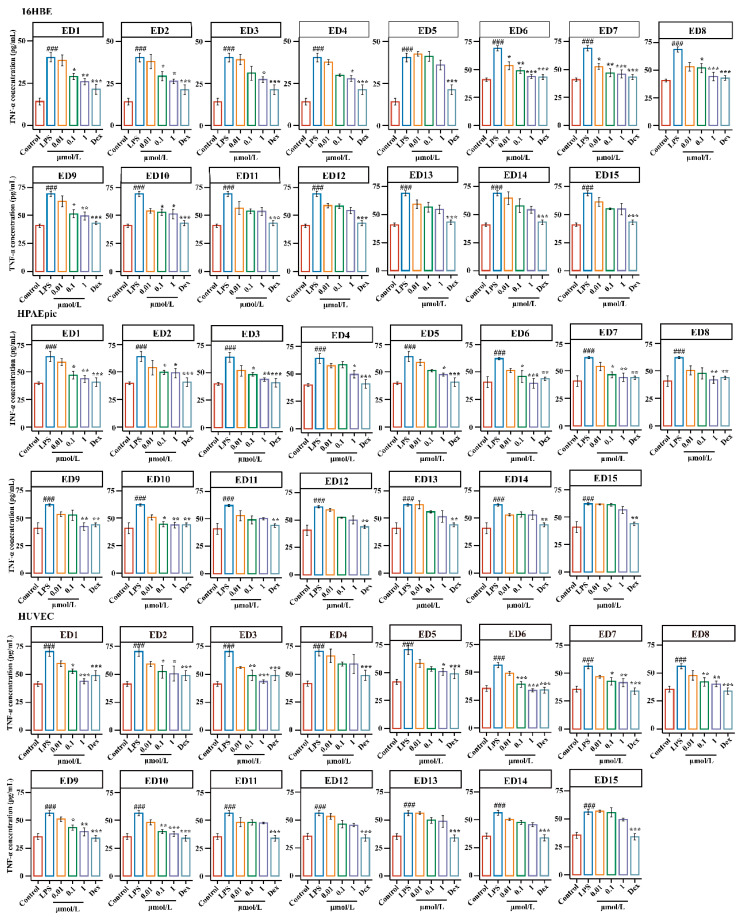
Effects of ED1-DE15 on TNF-*α* levels in LPS-induced cell injury in target cells. The 16HBE, HPAEpiCs and HUVECs were pretreated with ED1–15 at the concentration of 0.01–1 μM for 12 h. Dex was used as a positive control drug (5 μg/mL). TNF-*α* levels in the medium, as measured by ELISA. Results were obtained from three independent experiments and are presented as mean ± SEM. ^###^
*p* < 0.001 vs. Control; * *p* < 0.05, ** *p* < 0.01, *** *p* < 0.001 vs. LPS. LPS, Lipopolysaccharide; Dex, Dexamethasone.

**Figure 6 molecules-28-00295-f006:**
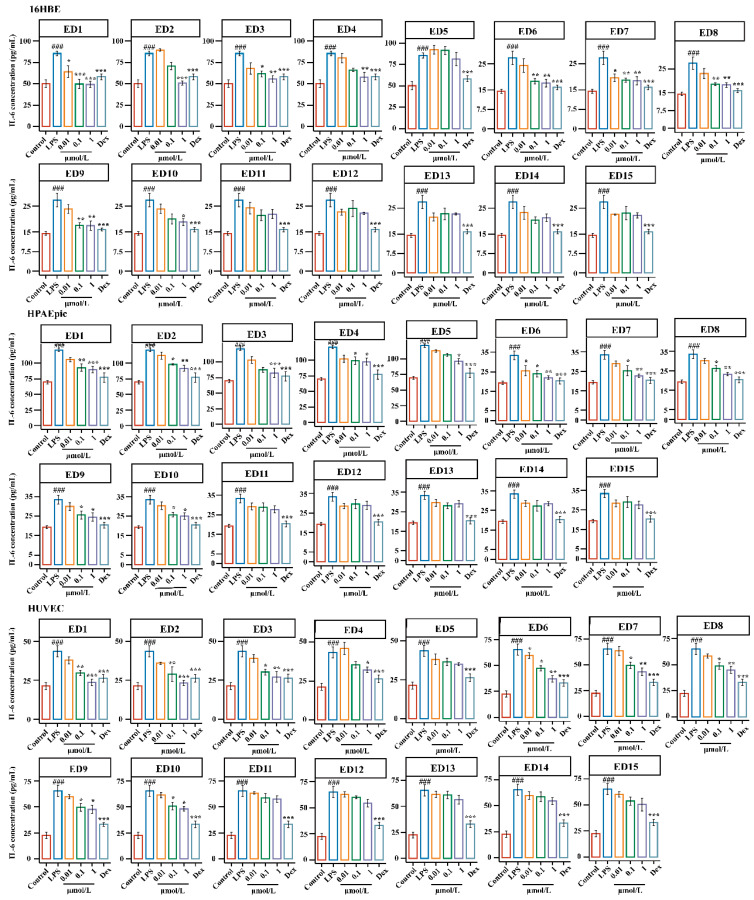
Effects of ED1-DE15 on IL-6 levels in LPS-induced cell injury in target cells. The 16HBE, HPAEpiCs and HUVECs were pretreated with ED1–15 at the concentration of 0.01–1 μM for 12 h. Dex was used as a positive control drug (5 μg/mL). IL-6 levels in the medium, as measured by ELISA. Results were obtained from three independent experiments and are presented as mean ± SEM. ^###^
*p* < 0.001 vs. Control; * *p* < 0.05, ** *p* < 0.01, *** *p* < 0.001 vs. LPS. LPS, Lipopolysaccharide; Dex, Dexamethasone.

**Figure 7 molecules-28-00295-f007:**
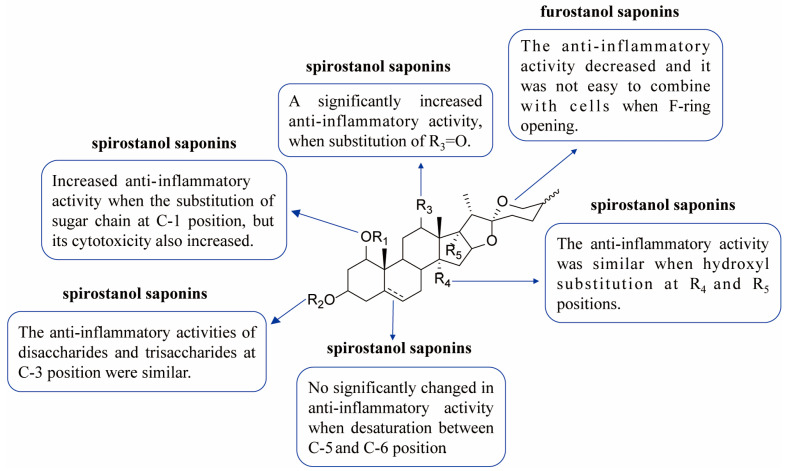
Structure–activity relationship involved in anti-inflammatory activity of spirostanol saponins and furostanol saponins in EDG.

**Table 1 molecules-28-00295-t001:** The compounds that bound to 16HBE, HPAEpiCs or HUVECs.

NO.	Compounds	16HBE	HPAEpiCs	HUVECs
1	26-*O*-*β*-D-glucopyranosyl-(25*R*)-5*β*-furostane-3*β*,22*α*,26-triol-3-*O*-*β*-D-xylopyranosyl-(1→4)-*β*-D-glucopyranosyl(1→2)-*β*-D-glucopyranoside	+	+	+
2	Ruscogenin 3-*O*-*α*-L-rhamnopyranosyl (1→2)-[*β*-D-glucopyr-anosyl(1→3)]-*O*-*β*-D-glucopyranoside	-	-	+
3	Ophiopogenin-3-*O*-*α*-L-rhamnpyranosyl-(1→2)-*β*-D-xylopyranosyl-(1→3)-*β*-D-glucopyranoside	-	-	+
4	Cixi-ophiopogon C	+	+	+
5	(25*R*)-3*β*-[(*O*-*β*-D-glucopyranosyl-(1→4)-*β*-D-galactopyranosyl)oxy]-5*β*-spirostan-12-one	+	+	+
6	Pennogenin 3-*O*-*α*-L-rhamnopyranosyl-(1→2)-*β*-D-xylopyra-nosyl-(1→4)-*β*-D-glucopyranoside	+	+	+
7	Prazerigenin A 3-*O*-*α*-L-rhamnopyranosyl-(1→2)-*β*-D-glucopyranoside	+	+	+
8	Ophiopogonanone E	+	+	+
9	Prazerigenin A-3-*O*-*α*-L-rhamnopyranosyl-(1→3)-*β*-D-xylopyranosyl-(1→4)-*β*-D-glucopyranoside	+	+	+
10	5,7,2’-Trihydroxy-3’,4’-methylenedioxy-6,8-dimethyl homoisoflavone	-	+	+
11	(25*S*)-3*β*-[(*O*-*α*-L-rhamnopyranosyl-(1→2)-*β*-D-galactopyranosyl)oxy]-5*β*-spirostan-12-one	+	+	+
12	3-*O*-*α*-L-rhamnopyranosyl-(1→2)-[*β*-D-glucopyranosyl(1→4)]-*β*-D-glucopyranosyl-(25*S*)-5*β*-spirostan-3*β*-ol	+	+	+
13	(25*S*)-5*β*-spirostan-3*β*-ol-3-*O*-*β*-d-glucopyranosyl-(1→2)-*β*-D-glucopy ranoside	+	+	+
14	Ophiopogonin D	+	+	+
15	Methylophiopogonone A	+	+	+
16	Ophiopogonin D’	+	+	+
17	Methylophiopogonone B	+	+	+
18	Methylophiopogonanone A	+	+	+
19	Methylophiopogonanone B	+	+	+
20	Ophiopogonin B	+	+	+

## Data Availability

Not applicable.

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
