# Peer review of "Screening and Identification of Anti-Inflammatory Compounds from Erdong Gao via Multiple-Target-Cell Extraction Coupled with HPLC-Q-TOF-MS/MS and Their Structure–Activity Relationship"

_molecules, 2022, doi:10.3390/molecules28010295_

Round 1

Reviewer 1 Report

Comments and Suggestions for Authors

This interesting study investigates the potential beneficial effects of active phytochemicals from Erdong Gao, a traditional Chinese formulation. The manuscript is well-written, scientifically substantial, and consistent and is in line with the journal's aims. This manuscript has merit and will interest readers; however, there are some drawbacks and many grammatical errors. I suggest the acceptance of the paper for publication after some minor revisions. Specifically:

The introduction section needs to contain information about two plants, the main components of Erdong Gao, a traditional Chinese formulation.

It would be nice to see in the manuscript an explanation of why these cell lines were taken for investigation and with which pulmonary problems are associated.

The selection of the two pro-inflammatory cytokines here investigated should be better justified in keeping with literature evidence, for example,

1.       Rübe CE, Wilfert F, Palm J, König J, Burdak-Rothkamm S, Liu L, Schuck A, Willich N, Rübe C. Irradiation induces a biphasic expression of pro-inflammatory cytokines in the lung. Strahlenther Onkol. 2004 Jul;180(7):442-8. doi: 10.1007/s00066-004-1265-7.

2.       Farzin H, Toroghi R, Haghparast A. Up-Regulation of Pro-Inflammatory Cytokines and Chemokine Production in Avian Influenza H9N2 Virus-Infected Human Lung Epithelial Cell Line (A549). Immunol Invest. 2016;45(2):116-29. doi: 10.3109/08820139.2015.1099663. Epub 2016 Feb 5.

3.       Yarmolinsky L, Budovsky A, Ben-Shabat S, Khalfin B, Gorelick J, Bishitz Y, Miloslavski R, Yarmolinsky L. Recent updates on the phytochemistry and pharmacological properties of Phlomis viscosa Poiret. Rejuvenation Research, 2018, Jan 8. doi: 10.1089/rej.2018.2093.

4.       Conti P, Ronconi G, Caraffa A, Gallenga CE, Ross R, Frydas I, Kritas SK. Induction of pro-inflammatory cytokines (IL-1 and IL-6) and lung inflammation by Coronavirus-19 (COVI-19 or SARS-CoV-2): anti-inflammatory strategies. J Biol Regul Homeost Agents. 2020;34(2):327-331. doi: 10.23812/CONTI-E.

In addition, the aim of the research should be shown clearly in the Introduction section.

The method section needs to contain information about the formulation preparation.

The mobile phase needs to be described correctly (Method section, part 2.4.1).

The discussion section should be improved by comprehensively analyzing the identified compounds and further studies.

Most abbreviations used at work have incomplete names. As it was mentioned, grammatical errors should be corrected. 

Author Response

Dear Editor and Reviewers,

Thank you and all reviewers very much for reviewing our manuscript (molecules-2075973) and giving us positive and constructive comments and suggestions. We have studied reviewer’s comments carefully and have made revision in the manuscript. We have tried our best to revise our manuscript according to the comments. Attached please find the revised version, which we would like to submit for your kind consideration.

Thank you very much for your kind consideration for further review and publication and I’m looking forward to hearing from you. If there are still some suggestions, we will do our best to improve and perfect it. 

Reviewer 2 Report

This manuscript deals with identifying the main anti-inflammatory phytochemicals in Erdong Gao (EDG), one of the traditional Asian medicines, and establishing their dose-response and structure activity relationships. To specifically distinguish those compounds that are potentially bioactive, the authors use human cells to ‘fish’ the potentially bioactives from the crude EDG extract, presuming binding -or cellular uptake- is the first and key step in exerting bioactivity. To verify bioactivity and determine dose-response and structure relationships of compounds that are commonly bound to three different cell types, they expose the same three cell types to increasing concentrations of each extracted phytochemical separately and check for effects on cell viability and LPS-induced inflammation responses. In principle, such approach using living cells is an interesting approach to filter potentially bioactive compounds out of the few thousand phytochemicals that can be present in a single crude plant extract. The anti-inflammatory activity of filtered (bound) compounds have been tested well and interesting structure-activity relationships were found in this manner. However, the present manuscript needs improvements at several places, including the use of English language: the current manuscript is hard to read and frequently difficult to understand due to many linguistic errors. Present rather than completed time should be used in cases of general statements, throughout manuscript (such as in line 35: ‘modulated’ should be ‘modulates’ and ‘was’ should be ‘is’; same errors in lines 32, 42, 43, 45, 50, 54, etcetera). The author should also discuss better their cell-based fishing method. In addition, since all data provided in this manuscript were generated using a single EDG water extract, the potential inconsistencies due to differences between extractions of the same EDG and between different EDG starting materials should be addressed. I also noticed some of the EDG compounds are incorrectly identified. Main comments and suggestions for improvements:

-       1st paragraph should be rephased to better describe the plant material and how is it extracted and applied as a medicin. Most non-Asian readers will not be familiar with EDG; its composition should be described with correct and official plant species names and tissue, e.g. roots of Asparagus cochinchinensis and Ophiopogon japonicus. The first record of EDG -according to the authors by She Sheng Mi Pou- is not a proper reference and no year of publication. Line 30: change “nourishing yin” into “nourishing the yin“.  

-       Line 43: “Hence it was necessary to look for the anti-inflammatory effective components in EDG”. Change “look for” into “investigate”. The authors should better describe why they believe their study is really necessary: around the world there are many TCM’s or other plants in use as medicines while their principal bioactive components are still unclear. I can agree if the authors mean it is necessary from a quality control point of view (stable efficacy between preparations).

-       Lines 59-60: Unclear, I guess the authors mean:  “Although this cell-based extraction method is rapid and convenient, it is still required to test and verify the bioactivity of the extracted compounds”. Actually, this is a critical point Here, and also in the discussion section, the authors should better discuss that such cell-based binding approach using crude complex plant extracts, such as EDG, unavoidably will not only isolate the aimed compounds bound to receptors / ion channels, but also many compounds that are differently (i.e. aspecifically) bound to the plasma membrane. Such a-specifically bound compounds are likely only partially removed by just repeated washing the cells in culture media or other solvents at neutral pH like PBS (line 166).  

-       The authors claim to have identified the components of EDG by LCMS (line 77), which actually were 124 phytochemicals (line 186). However, not all plant compounds ionize well in ESI-MS and it is also hard to believe that all EDG compounds can be separated by HPLC without co-elution. While 124 compounds seems a lot, it is a relative low number compared to what can nowadays detected in crude plant extracts using untargeted LCMS approaches. I would expect more compounds being present in EDG, especially in view of the fact that EDG consists of 2 different plant species. The likely possibility that the authors might have missed a number of EDG compounds should be discussed.

-       2.2: root material was dry, I presume. What were the temperatures used for root drying and subsequent extraction in water? Endogenous plant enzymes include various glycosidases that cause partial de-glycosylation of phytochemicals during extraction with water, if not inactivated by heating at drying or extraction. Actually, repeated extraction from the same EDG material, to check for reproducibility of the phytochemical composition of the water extract, has not been performed. Neither have different EDG preparations, e.g. from different harvest years, been compared in this study; e.g. how reproducible is Figure 1? Though this was not their aim, the authors should at least discuss in section 4 the inherent, natural variation in phytochemical composition between EDG preparations in relation to the observed bioactivities: all their data are based on a single EDG extract. I did not check literature, but I guess there will be data from studies towards variation between EDG materials, e.g. due to plant growth conditions, negetic diversity within both plant species, root drying method, etcetera.   

-       2.4.1 and 2.4.2 should be combined: the same HPLC system and conditions have been applied, only the detectors were different (ESLD and MS). Line 143: should be mentioned that both positive and negative ESI were used, in two separate LC runs. Were all indicated MS parameters the same in positive and negative mode? Line 147: what is meant with “sample collision energy”: collision cell energy?

-       Line 164-165: I guess the “Three cell lines were divided …..” are actually the three specific types mentioned at the beginning of line 164? Then please change line 165 into: “Cells of each line were divided …..”. What was the time of pretreatment? I seems this paragraph can be coupled to 2.3, as both involve the same cell types and methods, correct?

-       3.2 first paragraph should be moved to Introduction section, as it contains published work only.

-       Line 214: change “ …… capability and showed dose-dependent.” into “….. capability in a dose-dependent manner.” Again: unclear to me why it was “necessary” to screen for the anti-inflammatory compounds; do the authors mean “of our interest” or “interesting”?

-       Line 227: change into “……. in vitro, target cells were divided into …..” 

-       Lines 229-236. how can the authors be sure that the 15 compounds commonly extracted by the three cell types were indeed bound “specifically”? I do agree with the authors that both cell washing and using different cell types will help to remove at least a part of the a-specifically bound compounds. However, unless the authors have direct evidence that these 15 compounds commonly extracted after extensive cell washing are indeed all specifically bound to receptor proteins or are taken up by the cells, the use of the term ‘specifically’ is rather premature. As outlined above, a repetition of the entire cell-based binding experiment with preferably 3 independently prepared EDG extracts is needed to draw more solid conclusions.

-       Table 1: legend incorrect English

-       How sure are the authors about the provided structural details including stereometry of compounds mentioned in their manuscript (e.g. in 2.1., Figure 4, Table 1): have they all been unambiguously characterized using NMR? At least the identifications of compounds 1, 2 and 4 in Suppl Table S1 are incorrect: compound 1, observed mz 447.1661 cannot be C21H20O11: exact mass [M-H]- of C21H20O11 is 447.0933 and of C15H10O6 ([M-H-gluc]) 285.0405; EF should be C23H28O9 with M-gluc C17H18O4, thus this compound cannot be the indicated flavonoid. Same for compound 2: provided EF fits with neither the observed [M-H] nor the [M-H-gluc] mz; this compound cannot be the indicated flavonoid. Compound 4: calculated mz of C15H10O6 is 285.0406, which is too far off from the observed mz 285.1148; again not a flavonoid. By the way, if this compound was C15H10O6, how can the authors discriminate between luteolin and kaempferol sharing the same exact mass? The level of structural identification of all compounds should be provided in this table, e.g. according to the minimum reporting standards for chemical analyses (Sumner et al. Metabolomics 3, 2007). 

Author Response

(The authors gave the same response as above.)

Round 2

Reviewer 2 Report

The authors managed to improve their manuscript significantly, including English language, and provided proper answers to my remarks except for one: the wrong annotations of the compounds 1, 2 and 4. Although these compounds are not among the actual bioactives, it is key their annotations are correct as well in order to trust the annotations of the identified bioactive EDG compounds. The reply provided by the authors in the cover letter is insufficient and actually invalid. After incorporating some minor textual revisions (see below) and correcting the three wrong identifications, this manuscript shall be acceptable for publication. 

Line 8: change “Asparagi Radix and Ophiopogonis Radix” into species names and tissue: “roots of Asparagus cochinchinensis and Ophiopogon japonicus”

Line 29: change “was composed” into “is composed”

Line 37: change “such as” into “including”; Officinalisnin-II should be without capital (holds for all compound names in manuscript)

Line 39: typos: “Methyophiopogonone B ” should be “methylophiopogonone B”; display (plural)

Line 44: aquaporin-1 and aquaporin-5 (singular, and no capitals)

Line 76: methods (plural)

Line 86: change “allows direct application of HPLC-MS/MS technology after dissociation solution pretreatment to identify the target components online.” into “allows identification of the extracted compounds by HPLC-MS/MS”. Remove online, as the dissociation of the bound compounds is actually performed off line.   

Line 110:  ……fifteen compounds extracted in this manner ……..

Lines 124-134: Please verify all mentioned structures have previously been unambiguously identified by using NMR or authentic chemical standards, as accurate mass MS/MS alone is unable to structurally discriminate between the specified (stereo)isomeric forms like R vs S, alpha vs beta, etcetera.

Line 143: change power into powder;

Line 144: remove “extracting”. Please also indicate applied soaking and refluxing temperatures here. In their cover letter the authors reply EDG extraction is at 100 degrees C: so, the preparation was in total 7 hours at 100 degrees, correct? Btw: such long hot water exposure undoubtedly has a marked effect on the original metabolite composition of the root material (non-enzymatic oxidation, polymerization, deconjugation, etc). Any information available? Comparing the metabolite profile obtained with the currently applied hot water extraction procedure with that obtained by a direct 90% ethanol extraction of the EDG powder would help to determine the induced metabolite alterations and extraction efficiency of the traditional hot-water extraction procedure.     

Line 190: mass resolution used (FWHM)?

Line 262: change “Thus, it was necessary to screen for anti-inflammatory components in EDG.“ into “Thus, it was of our interest to identify the components underlying the observed anti-inflammatory activity of this EDG extract”

Line 286: Change into “Finally, fifteen components were found to bind to all three cell types and their chemical structures are shown in Figure 4.”

Line 310, legend of Figure 4: “The chemical structures of the compounds identified in the cell extract of EDG.” is confusing, as this figure actually shows those fifteen compounds commonly detected in the extracts of the three EDG-exposed human cells, rather than to the EDG extract itself. Please change legend accordingly.

Legend Table 1 should be “The compounds that bound to 16HBE, HPAEpic or HUVEC cells” Thus “or” instead of and, and “HUVEC cells” instead of HUVECs

Line 331: “…..were the potential anti-inflammatory compounds on EDG”.

Line 377: remove HUVEC here (but leave in line 379).

Line 413 and Suppl. Table S1: my remarks on incorrectly annotated compounds 1, 2 and 4 have not been considered well. The authors replied that the annotations of compounds 1, 2 and 4 are actually correct, just the observed masses of these specific molecules and fragments are wrong likely due to errors in instrument performance. However, this explanation is not valid, since even the in silico calculated masses provided by the authors do not fit with the provided elemental formula: e.g. for compound 1 both the observed m/z (447.1682) and the calculated m/z (447.1661) correspond within 5 ppm to C23H28O9, and certainly not to the indicated flavonoid compound C21H20O11 (with calculated m/z of 447.0933, i.e. 167 ppm off). Same holds for its MSMS fragment: observed m/z 285.1130 fits perfectly with C17H18O4 after loss of C6H10O5 (hexose, e.g. glucose) from the molecule, but definitely not to C15H10O6 (if kaempferol was the fragment). In addition, for all other compounds the observed m/z match well, with the provided elemental formula, while these were analyzed in the same LCMS run and with the same ionization settings. The clearly incorrect annotations of compounds 1, 2, and 4 must be corrected in the main text (e.g. line 413) and Table S1: sticking to the current wrong identifications would directly raise questions about the accuracy of the QTOF MS data and thus about the reliability of the identifications of all other compounds including the anti-inflammatory bioactives in EDG, which would be a shame. 

Author Response

 Dear Editor and Reviewers,

Thank you and all reviewers very much for reviewing our manuscript (molecules-2075973) and giving us positive and constructive comments and suggestions. We have studied reviewer’s comments carefully and have made revision in the manuscript. We have tried our best to revise our manuscript according to the comments. Attached please find the revised version, which we would like to submit for your kind consideration.

Thank you very much for your kind consideration for further review and publication and I’m looking forward to hearing from you. If there are still some suggestions, we will do our best to improve and perfect it.

Sincerely Yours.

BoYang Yu

Dr. & Professor

School of Traditional Chinese Pharmacy

China Pharmaceutical University (CPU)

639 Longmian Avenue, Jiangning District, Nanjing, 211198, P.R. China.

Tel: +86-025-8618-5157
